# Mechanochemical-Assisted Extraction and Hepatoprotective Activity Research of Flavonoids from Sea Buckthorn (*Hippophaë rhamnoides* L.) Pomaces

**DOI:** 10.3390/molecules26247615

**Published:** 2021-12-15

**Authors:** Zili Guo, Jingya Cheng, Lei Zheng, Wenhao Xu, Yuanyuan Xie

**Affiliations:** Collaborative Innovation Center of Yangtze River Delta Region Green Pharmaceuticals, College of Pharmaceutical Sciences, Zhejiang University of Technology, Hangzhou 310023, China; guozili@zjut.edu.cn (Z.G.); chengjy@zjut.edu.cn (J.C.); 18267735533@163.com (L.Z.)

**Keywords:** *Hippophaë rhamnoides* L., hepatoprotective activity, mechanochemical-assisted extraction, flavonoids, response surface methodology

## Abstract

Pomaces of sea buckthorn berry were usually side-products during the processing of juice. Due to a lack of an economical and effective extraction method, it was typically recognized as waste. For the purpose of resource utilization, the mechanochemical-assisted extraction (MCAE) method was applied to develop an ecofriendly extraction method and product with better pharmacology activity. The parameters were investigated through response surface methodology (RSM) design experiments. The processing conditions were optimized as follows: amount of Na_2_CO_3_ 40%, ball-to-material rate 29:1 g/g, milling speed 410 rpm, milling time 24 min, extraction temperature 25 °C, extraction time 20 min and the solid-to-solution ratio 1:10 g/mL. Under these conditions, the yields of flavonoids from sea buckthorn pomaces were 26.82 ± 0.53 mg/g, which corresponds to an increase of 2 times in comparison with that extracted by the heat reflux extraction method. Meanwhile, the hepatoprotective activity of sea buckthorn pomaces extracts was studied by the liver injury induced by *ip* injection of tetracycline. Biochemical and histopathological studies showed that biomarkers in serum and liver of nonalcoholic fatty liver disease (NAFLD) mice were significantly ameliorated when sea buckthorn flavonoids extracted by MCAE were used. Altogether, these results demonstrate that, as a green and efficient extraction, MCAE treatment could increase the extraction yield of sea buckthorn flavonoids, meanwhile it could exhibit significant activity of improving liver function. This research provided a new way to use pomaces of sea buckthorn as a functional food. It also has great value on the comprehensive utilization of nature’s resources.

## 1. Introduction

*Hippophaë rhamnoides* L. is a thorny small tree that is widely distributed throughout Asia and Europe. The fruit of *Hippophaë rhamnoides* L., known as “Shaji” (sea buckthorn), is a common eatable berry with great ecology and economic value around the world. Sea buckthorn is a kind of berry, which is also a rich source of nutraceuticals for human health. As a popular drinks, sea buckthorn juice is very popular in China, Russia and Eastern Europe [1]. However, after producing juice, a large number of pomaces contained high levels of total flavonoids and were discarded as waste or utilized rather inefficiently [2,3]. Previous research found that the peels, pulp and seeds of sea buckthorn contained different metabolites [4,5,6]. The whole berry and pulp extracts exhibited high flavonoid, carotenoid and polyphenol contents, whereas the seed portion expressed high tocopherol/tocotrienol and phytosterol concentrations [4]. The study also found that the peels and seed contained 65 metabolites (delphinidin, naringenin, pinocembrin, luteolin and taurine, etc.) and 130 metabolites (purines and nucleotides) that were absent in the juice. The juice contained exclusively 55 metabolites [5]. Recently, the total flavonoids from sea buckthorn had been reported as a functional food for treating cardiovascular diseases, which received recognition all over the world [7,8,9]. If high-value flavonoids with better pharmacological activity can be produced with high efficiency and low energy consumption extraction methods, the practical application of sea buckthorn pomaces can be broadened.

The traditional extraction methods of flavonoids include heat reflux extraction (HRE) [10], alkali extraction [11], ultrasound-assisted extraction [12,13], microwave-assisted extraction [14] and supercritical fluid extraction [15,16], etc. The processes of extracting flavonoids from sea buckthorn leaves mainly includes solvent extraction [17,18], ultrasonic-assisted extraction [19,20], microwave-assisted extraction [21], enzyme-assisted extraction [22] and synergistic extraction [23]. In recent years, the application of mechanochemistry as a preliminary treatment has represented a novel tool in the development of phytochemical manufacturing processes.

As an innovative mechanochemistry technique, MCAE has been developed and applied in the extraction of flavonoids from various natural plants, such as flavonoids from bamboo (*Phyllostachys edulis*) leaves [24], rutin from *Hibiscus mutabilis* L. [25], flavonoids from Ginkgo leaves [26], flavonoids from *Sophora flavescens* [27] and flavonoids from Chrysanthemum [28]. The above research showed that MCAE technology is very suitable for extracting flavonoids from various natural plants that are unstable for oxidative processes. This can reduce the extraction time and increase the extraction yield together. It is also useful to preserve the bio-activity of the active components, which are unstable due to oxidation.

From the previous research [23,24,25,26], MCAE had been applied to different plant materials to provide highly efficient extraction methods and plant extracts with better bio-activity. Thus, based on the experience, MCAE has been developed for the extraction of flavonoids from sea buckthorn pomaces to improve the conventional extraction methods and more effective extraction. The effects of the main operating parameters on the extraction yields of flavonoids from sea buckthorn pomaces were investigated. The MCAE method was compared with the HRE method. Moreover, hepatoprotective activity of total flavonoids obtained by MCAE was studied on the model of a fatty liver, which was induced by tetracycline in ICR mice.

## 2. Results and Discussion

### 2.1. Optimization of the MCAE Procedure

#### 2.1.1. Effects of Solid Reagent Type and Its Amount

Flavonoids contain lots of phenolic hydroxyl groups. They can be ionized with weak alkaline-like carbonate by mechanical force in solid states. After MCAE treatment, it can be dissolved in water quickly, extraction yield of flavonoids will be increased remarkably. In this experiment, different alkaline agents were chosen as a solid reagent under the following MCAE conditions: ball-to-material ratio 30:1 g/g, milling speed 400 rpm, milling time 20 min, extraction time 25 min, extraction temperature 25 °C and the ratio of solvent to solid 10:1 mL/g. Afterwards, the effect of the solid reagent amount (ranging from 10% to 50%) for flavonoids extraction from sea buckthorn was investigated. The relations between the two parameters and the flavonoids yield of sea buckthorn are displayed in Figure 1A,B. When milling with Na_2_CO_3_, the extraction yield is higher than that with other solid reagents. A possible reason is that Na_2_CO_3_ with better water solubility will be formed by alkaline additives and flavonoids under mechanical force. Meanwhile, the extraction yield of flavonoids increased rapidly with increasing the amount of solid reagent (from 10% to 40%). Conversely, after 40%, the increments of flavonoids yields were insignificant.

#### 2.1.2. Effects of MCAE Paraments

Ball-to-material ratio, milling speed and milling time determined energy output of MCAE, and they also affected the extraction yield directly. Following, these parameters were optimized individually.

The effect of the ball-to-material ratio on flavonoids’ extraction yields was determined under the following extraction conditions: the amount of solid reagent 40%, milling speed 400 rpm, milling time 20 min, extraction time 25 min, extraction temperature 25 °C, and solvent-to-solid ratio 10:1 mL/g. As shown in Figure 1C, when the ball-to-material ratio was 30:1 g/g, the flavonoids’ extraction yield was maintained at the maximum level. Thus, the ball-to-material ratio 30:1 g/g was selected as the optimum operating condition for MCAE.

The effect of ball milling speed on the flavonoid extraction was shown in Figure 1D. When the ball milling speed was set at 400 rpm, the flavonoids yield reached the maximum value, and there was no increase in the flavonoids yield thereafter. Theoretically, the MCAE of the ball mill can continuously destroy the cell wall and promote the reaction of biologically active substances with solid-phase reagents, thereby greatly improving the extraction efficiency. However, the ball milling speed at 500 rpm and 600 rpm did not further improve the entire extraction process. Therefore, a ball milling speed of 400 rpm was chosen for the rest of the follow-up experiment.

Milling times were investigated for 5, 10, 20, 30, 40, 50 and 60 min under the same extraction conditions. Total flavonoid content significantly increased from 12.34 ± 0.67 mg/g at 5 min to 26.73 ± 0.93 mg/g at 20 min and then slightly fell down, as shown in Figure 1E. MCAE pretreatment caused particle size reduction and the specific surface area increase, which makes the flavonoids in sea buckthorn have intimate contact with solid reagents. However, too long of a milling time could lead to a decrease in the recovery rate of flavonoids, which is mainly due to their oxidation and partial decomposition. Therefore, 20 min was selected as the optimum milling time.

#### 2.1.3. Effects of Extraction Process

On the basis of optimal milling conditions (ball-to-material ratio 30:1, milling speed 400 rpm, milling time 20 min, amount of Na_2_CO_3_ 40%), extraction conditions, including extraction time, extraction temperature and the solid-to-solution ratio, were investigated preliminary. Figure 1F showed the effect of the extraction temperature on the extraction yields of flavonoids. It was found that the yields of flavonoids decreased obviously, as the extraction temperature was raised from room temperature to 100 °C. The results showed that room temperature was an appropriate condition for the experiment. Figure 1G showed the effect of extraction time on the extraction yields of flavonoids. As the extraction time was extended from 5 to 20 min, a notable increase in the extraction rate was observed. The yields were decline when the extraction time was too long, and the optimal extraction time was 20 min. A possible reason is that the thermal stability of flavonoids is poor, and a high temperature will cause decomposition. Figure 1H showed the effect of the solid-to-solution ratio on the extraction yields of flavonoids. A significant increase in the extraction yields was observed with the solid-to-solution ratio raised from 1:5 g/mL to 1:10 g/mL. At a bigger solid-to-solution ratio, the yields kept at steady levels.

Through previous single-factor experiments, the ball-to-material ratio (20:1–50:1 g/g), milling speed (100–400 rpm) and milling time (10–60 min) were selected for RSM. Data were analyzed using design expert 7.1.6 software for statistical ANOVA, regression coefficients and regression equation. The polynomial equations, describing the yield of flavonoids (Y) as a simultaneous function of the milling speed (X_1_), milling time (X_2_) and ball-to-material ratio (X_3_), are shown in Equation (1).
Y = 26.73 + 0.09X_1_ + 1.88X_2_ − 0.45X_3_ + 0.50X_1_X_2_ + 0.19X_1_X_3_ + 0.51X_2_X_3_ − 1.34X_1_^2^ − 2.50X_2_^2^ − 2.39X_3_^2^(1)

In order to evaluate the optimal extraction conditions of MCAE for flavonoids and the relationship between the response and the significant variables, the model was analyzed by ANOVA. After fitting the experimental data (Appendix A Appendix A) to the quadratic polynomial model, as shown in Appendix A, the experimental data fitted well to the quadratic models. The analysis of variance of the response surface quadratic regression model showed that the model was highly significant (*p* < 0.0001) with a high F-value of 32.31.

Three-dimensional response surfaces using Equation (1) for the yield of flavonoids are shown in Figure 2. In order to describe the interactive influence of operational variables on responses, one variable remains unchanged, and the other two variables changed within a defined range. The shape of the response surfaces and contour plots indicated the nature and extent of the interaction between different variables. The regression analysis of the data showed the coefficient of determination (R^2^) values for flavonoids of 0.9765, which showed that the model was significant. The adjusted determination coefficient (Adj R^2^ = 0.9463) was also satisfactory to confirm the significance of the model. This showed that Equation (1) was suitable for describing the response of experiments related to flavonoids.

To determine the optimal level of the test variables for the yields of flavonoids, the 3D response surface described by the regression model is shown in Figure 2. By solving the inverse matrix, the optimal values of the variables affecting the yields of flavonoids given by the software were milling speed 410.13 rpm, milling time 23.81 min and ball-to-material rate 29.51:1. The model gave the maximum predicted values of flavonoids (27.10 mg/g) under these optimal conditions, slightly higher than 26.82 mg/g obtained from the plot analysis. Considering the operating convenience, the optimal extraction parameters were determined to be milling speed 410 rpm, milling time 24 min and ball-to-material rate 29:1.

Triplicate experiments were carried out under the determined conditions, and the yields (26.82 ± 0.53 mg/g) consistent with the predicted value (27.10 mg/g) were obtained, indicating that the model was suitable for the extraction process of sea buckthorn.

In conclusion, under optimized conditions, the MCAE method to extract sea buckthorn flavonoids showed significant advantages on the extraction yield and energy consumption. Compared with the HRE method, the extraction time of MCAE was shortened to only 20% of the HRE method’s extraction time, as shown in Table 1. Moreover, the MCAE method adopted water as a solvent, which was much safer and greener. Therefore, MCAE is an efficient and environmentally friendly alternative to utilize for sea buckthorn pomaces.

### 2.2. Quantitative and Morphology Analyses

Quantitative and morphology analyses were carried out by HPLC and SEM to demonstrate the advantages of the MCAE method and explain possible mechanisms in this part.

A reverse-phase high-performance liquid chromatography (HPLC) method was developed for the quantitative analyses of its major constituents rutin, quercetin, and isorhamnetin. In order to develop effective mobile phases, various solvent systems were tested, including acetonitrile, methanol and different combinations of water with phosphoric acid. Finally, a solvent system consisting of 0.4% phosphoric acid in water and methanol was proved to be successful. As shown in Appendix A, the retention times of rutin, quercetin and isorhamnetin were 19.317 min, 20.331 min and 21.005 min, respectively. Appendix A is the HPLC diagram of sea buckthorn flavonoids extracted by the HRE method (HPG). Appendix A is the HPLC diagram of sea buckthorn flavonoids extracted by the MCAE method (MPG). From those HPLC diagrams, according to the content calculation formula, the concentrations of rutin (925.14 μg/mL), quercetin (1091.1 μg/mL) and isorhamnetin (433.86 μg/mL) in MPG were obviously higher than HPG, especially those insoluble flavonoids (quercetin, and isorhamnetin). This result fully expressed the advantage of the MCAE method on sea buckthorn flavonoids’ extraction.

Furthermore, morphology analyses of sea buckthorn samples treated before and after MCAE are shown in the following. Figure 3A,B shows the micrograph of raw sea buckthorn powder, which has a bigger particle size, an unbroken cell structure and highly rough surfaces. However, after MCAE treatment (Figure 3C,D), the particle size of sea buckthorn samples was obviously reduced, and the cell structure of sea buckthorn was almost destroyed. Thus, water was efficiently permeated into the cell wall, and the extraction yield significantly increased simultaneously.

### 2.3. Pharmacology Study

Sea buckthorn flavonoids had been reported as a functional food for NAFLD. Herein, NAFLD mouse model was applied to compare liver protection effect between MPG and HPG. Through body weight, liver index, serum index and histopathological studies, excellent liver protective activity of MPG was fully demonstrated.

#### 2.3.1. Body Weight

The body weights of the mice were recorded every day. As shown in Figure 4A, the body weights of mice in the normal control group (NCG), NAFLD model group (NMG), MPG group (MPG), HPG group (HPG) and curcumin control group (CCG) all showed an increasing trend. The initial weights of NMG were slightly lower than that of the normal group.

#### 2.3.2. Liver Index

The ratio of wet liver weight to the body weight of the mouse reflects the accumulation of lipids in the liver. The liver index of the NMG was increased by 38.80% compared to the normal control group (*** *p* < 0.005). NMG, MPG, HPG and CCG decreased by 29.85%, 29.13% and 36.40%, respectively (*** *p* < 0.005, *** *p* < 0.005, **** *p* < 0.001). The liver index of the mice is shown in Figure 4B.

#### 2.3.3. Serum Index

The serum triglycerides (TG), total cholesterol (TC), low-density lipoprotein (LDL-C), high-density lipoprotein (HDL-C), aspartate aminotransferase (AST) and alanine aminotransferase (ALT) of NMG significantly increased compared with NCG. Compared with NMG, the serum TG, TC, LDL-C, AST and ALT of MPG, HPG and CCG significantly reduced. The serum index experiment results are shown in Figure 4C–H (* *p* < 0.05; ** *p* < 0.01; *** *p* < 0.005; **** *p* < 0.001).

#### 2.3.4. Liver Index

As shown in Appendix A and Figure 4I–L, the liver TG, TC and LDC-C of NMG significantly increased compared with NCG, while the HDL-C significantly decreased. The liver TG of MPG and CCG were significantly reduced compared with NMG. The liver TG of HPG was reduced. The liver TC and LDC-C of MPG significantly reduced, while the liver TC and LDC-C of HPG and CCG reduced. The HDL-C of MPG, HPG and CCG significantly increased (* *p* < 0.05; ** *p* < 0.01; *** *p* < 0.005; **** *p* < 0.001).

From the above results, it can be seen that MPG has a significant therapeutic effect on fatty liver in NAFLD mice, and it has a better effect than curcumin.

#### 2.3.5. Histopathological Studies

The results of the histopathological examination are shown in Figure 5. Histopathological studies of NCG liver sections showed regular cell structures with different hepatocytes, sinusoidal spaces and central veins (Figure 5A). Hepatocytes were polygonal cells with retained cytoplasm and obvious nuclei. On the other hand, in the NMG, the histological examination showed structural loss, inflammation and congestion with cytoplasmic vacuolation, fat changes, sinusoidal dilation and necrosis of the lobule center. It also showed collagen bundles around the lobules, resulting in huge fibrous septa and distorted tissue structure (Figure 5B). Animals treated with the MPG showed that the hepatic cords were arranged neatly, no obvious expansion or squeezing of the liver sinusoids and no obvious inflammation; a large number of hepatocytes were widely seen to be cytoplasmic vacuolation (Figure 5C). Animals treated with the HPG showed that the hepatic cords were arranged neatly, and the liver sinusoids were not significantly expanded or squeezed; more hepatocytes were seen in the tissue, and the cytoplasm was slightly loose; lymphocytes and neutrophil infiltration were rare (Figure 5D). Histopathological studies also showed better recovery of NAFLD by flavonoids from sea buckthorn as compared to curcumin (Figure 5E).

From the above results, it can be seen that MPG has a significant therapeutic effect on fatty liver in NAFLD mice, better than HPG even better than curcumin. This result shows that besides enhancing the extraction yield, the MCAE method also raised the pharmacology activities of sea buckthorn flavonoids.

These above results were consistent with the results reported in the literature that rutin and quercetin exhibited hepatoprotective effects [29,30,31,32,33,34,35,36,37]. Liu et al. [29] tried to investigate the molecular mechanisms underlying rutin’s hypolipidemic and hepatoprotective effects in NAFLD. The experimental data demonstrated that rutin could reduce TG content and mitigate oxidative injuries in fat-enriched hepatocytes. Pan et al. [30] showed that dietary rutin supplementation had a beneficial effect against cholestatic liver injury, as evidenced by the alleviation of histopathological changes and the improvement of serum bio-chemicals, such as AST, ALT, total bilirubin, TG, TC and total bile acids, using a rat bile duct ligation model. Domitrovic et al. [31] investigated the mechanisms underlying the protective effects of rutin and its aglycone quercetin against CCl4-induced liver damage in mice. The results demonstrated that rutin and quercetin could ameliorate acute liver damage by at least four mechanisms: acting as scavengers of free radicals, inhibiting NF-κB activation and the inflammatory response, exerting antifibrotic potential and inducing the Nrf2/HO-1 pathway. The rutinoside moiety in position 3 of the C ring could be responsible for more pronounced protective effects against iNOS induction, nitrosative stress and hepatocellular necrosis. Miltonprabu et al. [32] analyzed the available literature regarding the hepatoprotective effects of quercetin with special emphasis on its mechanisms of action. The data showed that quercetin appears to be a promising hepatoprotective compound. Sotiropoulou et al. [33] clearly demonstrated that quercetin had potent antioxidative stress action and inhibitory effects on hepatocyte apoptosis, inflammation, and generation of reactive oxygen species, factors which were linked to the development of the disease. Ying et al. [34] also showed that oral administration of quercetin at doses of 30–60 mg/kg to hyperlipidemic rats for 14 days was highly effective in decreasing the levels of serum TC, TG, LDL-C, ALT and AST. Fuentes et al. [35,36] also investigated the potential of the quercetin oxidation metabolite to protect Caco-2 monolayers against oxidative stress (OS) and the loss of the intestinal epithelial barrier function (IEBF) in Caco-2 cell monolayers.

## 3. Materials and Methods

### 3.1. Plant Materials and Chemicals

The fresh fruits of sea buckthorn used in this experiment were provided and identified by Qinghai Nationalities University. They were stored in the −18 °C freezer until use.

Reference substances of rutin (purity ≥ 98%), quercetin (purity ≥ 98%), isorhamnetin (purity ≥ 98%) were purchased from Shanghai yuanye Bio-Technology Co., Ltd. (Shanghai, China). Ultrapure water was produced by Barnstead TII super Pure Water System (MA, USA). D101 macroporous resin was purchased from Tianjin Bohong Resin Technology Co., Ltd. (Tianjin, China). All other analytical grade chemicals used in this experiment were purchased from Sinopharm Chemical Reagent Co., Ltd. (Shanghai, China).

High-fat feed was purchased from SPF (Beijing, China) Biotechnology Co., Ltd. (Beijing, China). All detection kits (total cholesterol test kit, triglyceride detection kit, protein carbonyl detection kit, high-density lipoprotein cholesterol detection kit, low-density lipoprotein cholesterol detection kit, aspartate aminotransferase detection kit, alanine transaminase detection kit and Bradford protein detection kit) were purchased from Nanjing Jiancheng Bioengineering Research Institute (Nanjing, Jiangsu, China).

### 3.2. Pretreatment and Mechanochemical-Assisted Extraction (MCAE)

Fresh fruits of sea buckthorn were juiced by a single-screw press, and residues were collected and fully dried below 5% water contained. Then, sea buckthorn residue was defatted with petroleum ether and dried to constant weight at 50 °C.

Defatted sea buckthorn pomaces (5.0 g) and different amounts of solid reagent (10–50%) were added into PM400 high-intensity planetary activator (grinding media: stainless steel balls of 8 mm diameter; weight of balls: 4.2 g; two drums and 50 mL each; the volume of load/drum ratio: 1:2). After milling for several minutes (5–60 min), the powder was extracted with an appropriate volume of water (1:5–1:60 mL/g) with a certain temperature (25–100 °C) and time (5–120 min). Then the mixture was clarified by centrifugation at 2000 rpm for 10 min. The supernatant was adjusted to pH 6.0 with 10% hydrochloric acid. Lastly, the liquid product was condensed and centrifuged at 2000 rpm for 10 min to collect crude total flavonoids and analyzed by UV and HPLC analysis. The analysis and further process are shown in Figure 6. The technological parameters, such as solid reagent type and its amount, ball-to-material ratio, milling speed, milling time, extraction temperature, extraction time, and the solid-to-solution ratio, were optimized in terms of flavonoids yield.

### 3.3. Total Flavonoids Determination

Total flavonoids content was determined by the NaNO_2_–Al(NO_3_)_3_–NaOH colorimetry method. The resulting red aluminum complex was then measured at 510 nm. Rutin was used as a standard for the calibration curve. The total flavonoid content was calibrated using the linear equation based on the calibration curve.

### 3.4. Purification of Total Flavonoids and HPLC Analysis

The crude extract of total flavonoids extracted by MCAE was dissolved in water. The concentrated filtrate was loaded with a peristaltic pump using a wet packing column at a rate of 0.5 BV/h. Stay and adsorb for 8 h, then sequentially eluted with 1 BV purified water, 2 BV 30% ethanol and 2 BV 60% ethanol at a speed of 1 BV/h, respectively. Each gradient eluate was collected, concentrated, freeze-dried and named S0, S30, S60.

HPLC analysis was used to determine the flavonoid (rutin, quercetin and isorhamnetin) content in the sea buckthorn pomaces extracts. The analytical separation was performed on an Agilent 1100 HPLC (Agilent, Santa Clara, CA, USA) with a Welchrom^®^ C18 column (4.6 mm × 250 mm, 5 μm, Welch, China). Solvent A (0.4% phosphoric acid) and solvent B (methanol) were selected as the mobile phases. Gradient elution was used as follows: 0–10 min, 15% B; 10–15 min, 15–85% B; 15–25 min, 85% B; 25–30 min, 85–15% B; 30–35 min, 15%. The injection volume was 10 μL, the flow rate was 1.0 mL/min and the column temperature was maintained at 25 °C. The signal was monitored at 254 nm.

The peaks were characterized by comparing the retention time with the standards. The internal standard method was used to calculate the content of each compound with the IS of para-aminobenzoic acid. The standard solutions were prepared as follows: rutin (50 μg/mL), quercetin (50 μg/mL), isorhamnetin (50 μg/mL) and para-aminobenzoic acid (20 μg/mL). The content calculation formula was as follows:$$\mathrm{Correction}\;\mathrm{factor}\;f=\frac{As/ms}{Ar/mr}$$

*As*: The peak area of IS; *Ar*: The peak area of standard; *ms*: The amount of IS added; *mr*: The amount of standard added.
$$\mathrm{Sample}\;\mathrm{content}\;m_{i}=f\times \frac{Ai}{As/ms}$$

*Ai:* The peak area of analyte; *As:* The peak area of IS; *ms:* The amount of IS added.

### 3.5. Scanning Electron Microscopy (SEM)

The morphological alterations of raw samples and obtained from the MCAE method were observed by SEM using ZEISS Gemini 500 field SEM (Carl Zeiss AG, Jena, Germany).

### 3.6. Animals

From the Zhejiang Academy of Medical Sciences (Hangzhou, China), 40 SPF-grade male ICR mice (5–6 weeks old) were kept at an animal room (ambient temperature of 22 ± 2 °C, relative humidity of 55 ± 5%, 12 h light/dark cycles). They were observed for one week in the Experimental Animal Center of the Zhejiang Province (Hangzhou, China) before starting the experiments. They were fed with freely available food and water and fasted with free access to water for 12 h before drug administration. The experimental protocols involving animals strictly followed the Guide for the Institutional Animal Care and Use Committee of Zhejiang University of Technology Laboratory Animal Center (20190927083). The animals were randomly divided into 5 groups of 8 mice each and treated as follows.

Group A (NCG): ordinary feed + saline (*ip*) for 5 days + purified water for 15 days.

Group B (NMG): high-fat diet + tetracycline saline (150 mg/kg, *ip*) at 10 am daily for 5 days + purified water for 15 days.

Group C (MPG): high-fat diet + tetracycline saline (150 mg/kg, *ip*) for 5 days + MCAE flavonoids (200 mg/kg, *po*) for 15 days.

Group D (HPG): high-fat diet + tetracycline saline (150 mg/kg, *ip*) for 5 days + HRE flavonoids (200 mg/kg, *po*) for 15 days.

Group E (CCG): high-fat diet + tetracycline saline (150 mg/kg, *ip*) for 5 days + curcumin (200 mg/kg, *po*) for 15 days.

The body weights of all mice were measured every day. All mice were sacrificed 12 h after the last treatment and overnight fast. Blood samples were collected, and serum was separated for assay of the liver biomarker. The liver was harvested, washed with physiological saline and blotted dry with filter paper, then weighed. Gross examination was conducted to examine any abnormalities developed in the organs. Subsequently, the livers of all mice were subjected to histopathological examination in a blinded manner.

### 3.7. Biochemical Examination and Histopathological Studies

The collected blood samples were separated at 3000 rpm for 10 min. TC, TG, HDL-C, LDL-C, AST, ALT and liver biochemical indicators were assayed by standard automated techniques according to the procedures [38].

The liver was sliced, and pieces were fixed in 10% buffered formaldehyde solution for histological study. The fixed tissues were processed by an automated tissue processing machine. Tissues were embedded in paraffin wax by conventional methods. Sections of 4 µm in thickness were prepared and then stained with hematoxylin eosin [39]. Afterwards, the sections were observed under the microscope for histopathological changes, and their photomicrographs were captured.

### 3.8. Statistical Analysis

All values were expressed as mean ± SEM. Statistical analysis was performed using Prisma 6 software (PRISMA Technology Inc., Chicago, IL, USA). Data were evaluated for significance one-way analysis of variance (ANOVA) followed by Bonferroni’s multiple comparison test. A value of *p* < 0.05 was considered to be significant.

## 4. Conclusions

In this study, the MCAE technology was used to extract flavonoids from sea buckthorn pomaces. The optimal operating parameters were determined by response surface methodology design experiments. Under these optimal conditions, the experimental value agreed with the predicted value, indicating the success of RSM for optimizing flavonoids from sea buckthorn pomaces. The sea buckthorn flavonoids extracted by the MCAE method performed significant hepatoprotective activity that was proven by biochemical and histopathological analyses. Based on the results, MCAE has notable advantages of reducing organic solvent, saving time and using a low temperature with higher efficiency; thus, the method represents a valuable alternative to the traditional HRE for the preparation of flavonoids. Meanwhile, the present investigation indicates flavonoids extracted by MCAE may be more effective than flavonoids extracted by HRE for the treatment of NAFLD. A possible mechanism was related to a higher concentration of different flavonoids in the extract. In conclusion, this study provided a novel method for utilizing sea buckthorn pomaces in an economical and eco-friendly way. Meanwhile, it will also be very valuable for the development of sea buckthorn flavonoids’ health food and products.

## Figures and Tables

**Figure 1 molecules-26-07615-f001:**
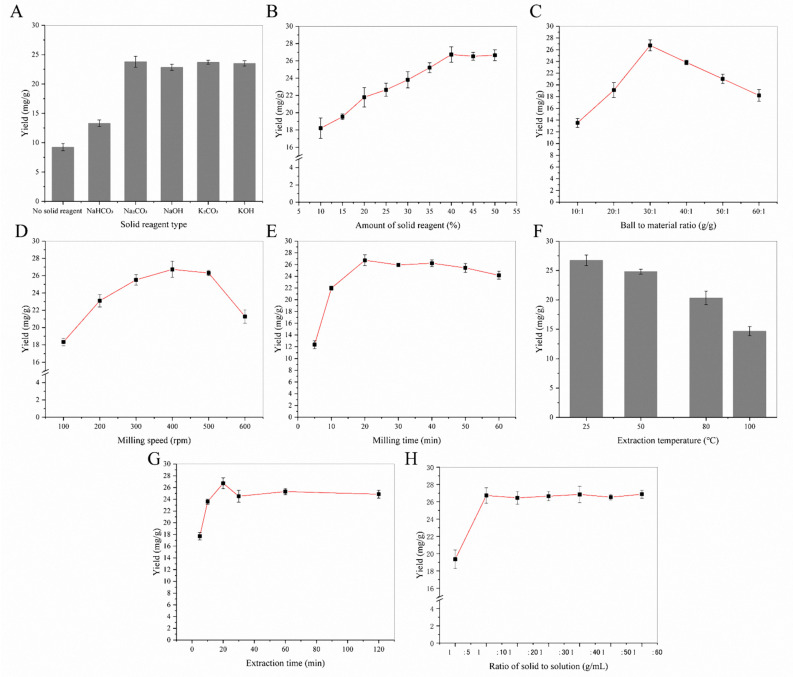
The single-factor experiment of MCAE: (**A**) the effect of solid reagent type; (**B**) the effect of solid reagent amount; (**C**) the effect of ball-to-material ratio; (**D**) the effect of milling speed; (**E**) the effect of milling time; (**F**) the effect of extraction temperature; (**G**) the effect of extraction time; (**H**) the effect of the solid-to-solution ratio.

**Figure 2 molecules-26-07615-f002:**
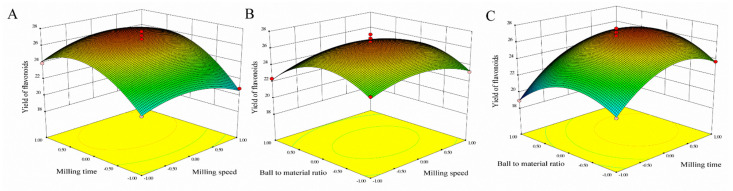
Response surfaces representations for MCAE operating parameters. (**A**) Varying milling speed and milling time; (**B**) Varying ball-to-material and milling speed; (**C**) Varying ball-to-material and milling time.

**Figure 3 molecules-26-07615-f003:**
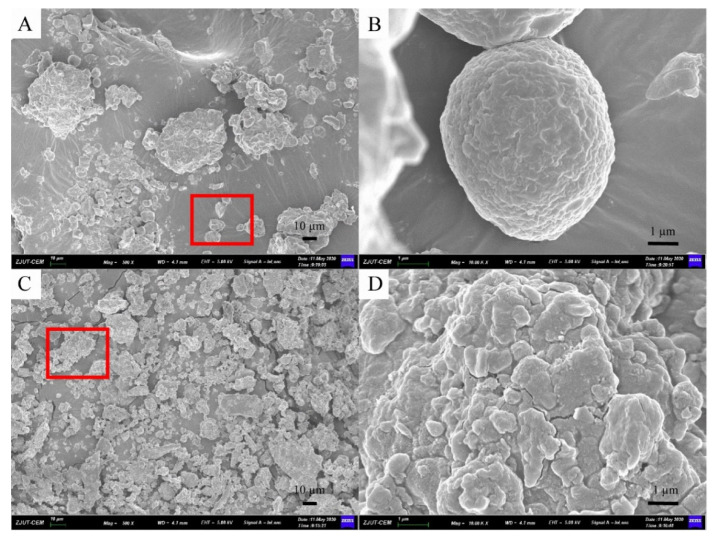
The SEM micrograph of: (**A**) sea buckthorn powder of raw material (500×); (**B**) figure of red area in A (10.00K×); (**C**) milled with Na_2_CO_3_ for 20 min (500×); (**D**) figure of red area in (**C**) (10.00K×).

**Figure 4 molecules-26-07615-f004:**
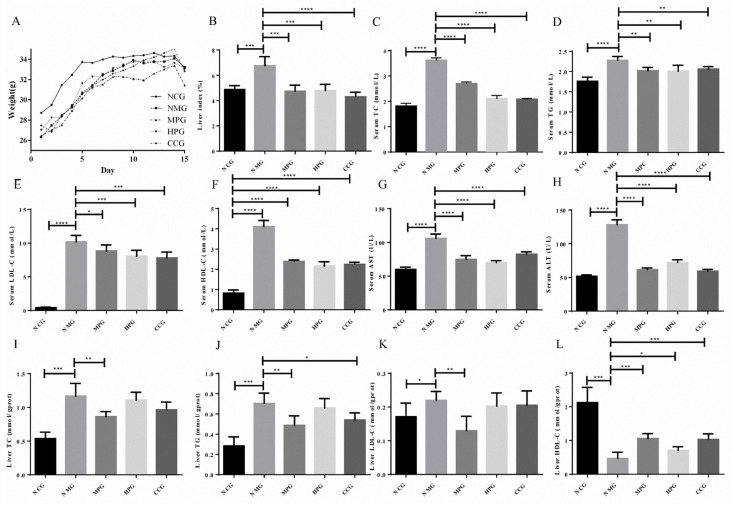
(**A**) the body weight of the mice; (**B**) liver index of the mice; (**C**) serum TC index results; (**D**) serum TG index results; (**E**) serum LDL-C index results; (**F**) serum HDL-C index results; (**G**) serum AST index results; (**H**) serum ALT index results; (**I**) liver TC index results; (**J**) liver TG index results; (**K**) liver LDL-C index results; (**L**) liver HDL-C index results. * There is difference; ** The difference is obvious; *** The difference is high obvious; **** The difference is very high obvious.

**Figure 5 molecules-26-07615-f005:**
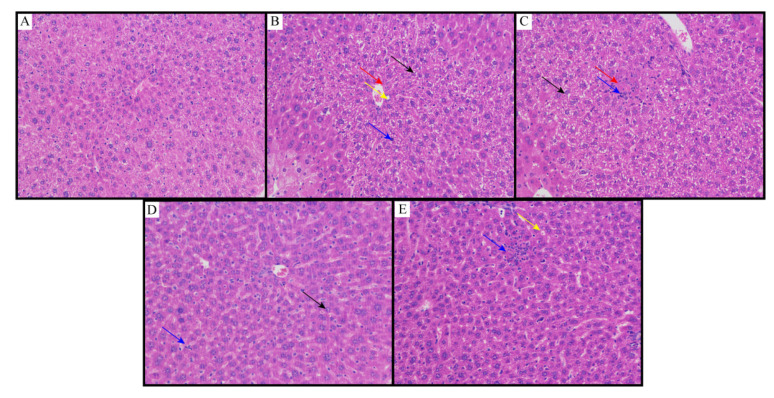
Liver slices of (**A**) NCG; (**B**) NMG; (**C**) MPG; (**D**) HPG; (**E**) CCG. Eight animals of each group were investigated (proliferative connective tissue: red arrow; loose cytoplasm: black arrow; cytoplasmic vacuolation: yellow arrow; lymphocytes and neutrophil infiltration: blue arrow).

**Figure 6 molecules-26-07615-f006:**
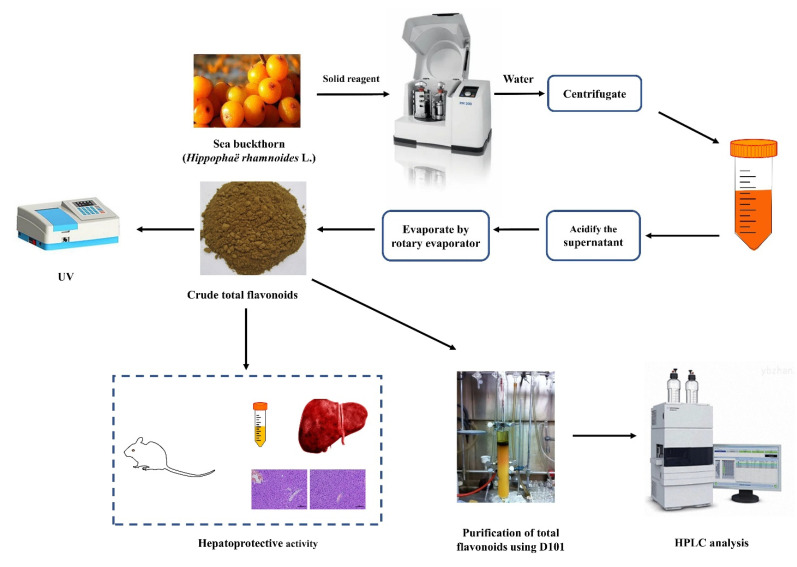
Scheme of MCAE method and further process.

**Table 1 molecules-26-07615-t001:** Comparison of MCAE with HRE methods.

Extraction Methods	Extraction Time	Extraction Temperature	Solvent	Amount of Solvent	Yield of Flavonoids (mg/g) ^a^
MCAE	24 min	25 °C	Water	1:10 g/mL	26.82 ± 0.53
HRE	2 h	50 °C	Ethanol	1:40 g/mL	8.99 ± 0.10

^a^ Data are presented as means ± SD (*n* = 3).

## Data Availability

Not applicable.

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
