# Peer review of "Mechanochemical-Assisted Extraction and Hepatoprotective Activity Research of Flavonoids from Sea Buckthorn (Hippophaë rhamnoides L.) Pomaces"

_molecules, 2021, doi:10.3390/molecules26247615_

Round 1

Reviewer 1 Report

This manuscript investigated the “Mechanochemical-assisted extraction and hepatoprotective activity research of flavonoids from sea buckthorn (Hippophae rhamnoides L.) pomaces”. The following comments have been outlined to improve the quality.

  • There several typographical error, check the manuscript careful to fix them.
  • Remove pronouns e.g. we, our, etc from the manuscript and rewrite the concerned sentences accordingly.

Author Response

Dear reviewer,

We would like to thank you for the careful reading of our manuscript (molecules-1460559) entitled “Mechanochemical-assisted extraction and hepatoprotective activity research of flavonoids from sea buckthorn (Hippophaë rhamnoides L.) pomaces”. According to the comments, we have carefully revised the manuscript and found these comments are helpful in improving the quality of the manuscript. Listed below are the reviewers' comments and the changes that were made in the manuscript.

Reviewer 2 Report

This work deals with the extraction and hepatoprotective activity of flavonoids from the pomace of H. rhamnoides.

The pomace of sea buckthorn obviously contains remnants of the peel, pulp and also seeds. All these components differ considerably in their content of compounds. Especially the seeds are rich in oleaginous substances. Have you please tried to distinguish between these fractions in terms of flavonoid content. Can you comment on this issue in the paper?

The "Materials" section is completely missing from the paper - you need to provide the source, composition and characterization of the raw material used in your study.

You should better describe the phytochemical parameters of sea buckthorn in the introduction. How variable is the composition of your material?

Since the major flavonoid components of your extract are rutin and quercetin, it would be extremely helpful to use these pure substances as (positive) controls in the hepatoprotective assays. This would give a clear idea of how the crude extract differs from pure and defined compounds, which are also used in various cytoprotective applications anyway.

It is highly advisable to add to the list of abbreviations at the end of the article.

The references are not properly formatted, the format of the references is inappropriate - please refer to the instructions or read a more recent paper in the journal. DOI must be added to all references. Italics must be used for Latin names - e.g. ref. 11 (and not only references), some references contain typos, e.g. ref. 15 and some others.

Author Response

(The authors gave the same response as above.)

Round 2

Reviewer 2 Report

The authors responded to the referee's report. Regrettably, only 3 issues (out of 6) were addressed, although not properly - references: there must be full stops in the journal abbreviations (missing now), but NOT at the end of the reference (read the instructions for authors carefully). Materials and abbreviations are OK.

All other points have been virtually unaddressed by the authors, which is unacceptable. 

Q#1 - Content of comments on the different fractions of the pomace in the paper, literature data are acceptable.

Q#4 -I understand you refuse to do further experiments with rutin and quercetin, which I am not happy about. Anyway, then you need to search the literature and find relevant data on these two compounds on the respective molecular models (cell lines used in you study) with these compounds. If these data are not available, then you need to do these experiments with these pure compounds.

This journal is about MOLECULES and therefore about molecular effects. Your topic without these experiments would then be better suited for a biotechnology journal.

Therefore, I cannot recommend your paper for publication now.

Author Response

Dear reviewer,

We really appreciate your consideration for giving us the opportunity to resubmit our manuscript (molecules-1460559) entitled “Mechanochemical-assisted extraction and hepatoprotective activity research of flavonoids from sea buckthorn (Hippophaë rhamnoides L.) pomaces”. First of all, we are sorry for that the response was not satisfactory to you.

Based on your comments, we carefully revised the manuscript again, and hoped that this reply could be accepted. Listed below are the comments and the changes that were made in the manuscript.
